# Retardation of Bacterial Biofilm Formation by Coating Urinary Catheters with Metal Nanoparticle-Stabilized Polymers

**DOI:** 10.3390/microorganisms10071297

**Published:** 2022-06-27

**Authors:** Osamah Al Rugaie, Ahmed A. H. Abdellatif, Mohamed A. El-Mokhtar, Marwa A. Sabet, Ahmed Abdelfattah, Mansour Alsharidah, Musaed Aldubaib, Hassan Barakat, Suha Mujahed Abudoleh, Khalid A. Al-Regaiey, Hesham M. Tawfeek

**Affiliations:** 1Department of Basic Medical Sciences, College of Medicine and Medical Sciences, Qassim University, P.O. Box 991, Unaizah 51911, Saudi Arabia; 2Department of Pharmaceutics, College of Pharmacy, Qassim University, Buraydah 51452, Saudi Arabia; a.abdellatif@qu.edu.sa; 3Department of Pharmaceutics and Pharmaceutical Technology, Faculty of Pharmacy, Al-Azhar University, Assiut 71524, Egypt; 4Department of Medical Microbiology and Immunology, Faculty of Medicine, Assiut University, Assiut 71515, Egypt; elmokhtarma@aun.edu.eg; 5Department of Microbiology and Immunology, Faculty of Pharmacy, Sphinx University, New-Assiut 71684, Egypt; marwasabet@gmail.com; 6Department of Industrial Pharmacy, Faculty of Pharmacy, Assiut University, Assiut 71526, Egypt; ahmed.abdelafattah@pharm.aun.edu.eg; 7Department of Physiology, College of Medicine, Qassim University, Buraydah 51452, Saudi Arabia; malsharidah@qu.edu.sa; 8Department of Veterinary Medicine, College of Agriculture and Veterinary Medicine, Qassim University, Buraydah 51911, Saudi Arabia; drmusaad@qu.edu.sa; 9Department of Food Science and Human Nutrition, College of Agriculture and Veterinary Medicine, Qassim University, Buraydah 51452, Saudi Arabia; haa.mohamed@qu.edu.sa; 10Food Technology Department, Faculty of Agriculture, Benha University, Moshtohor 13736, Egypt; 11Department of Basic Pharmaceutical Sciences, Faculty of Pharmacy, Isra University, Amman 11622, Jordan; suha.abudoleh@iu.edu.jo; 12Department of Physiology, College of Medicine, King Saud University, Riyadh 11451, Saudi Arabia; kalregaiey@ksu.edu.sa

**Keywords:** silver nanoparticles, urinary catheters, biofilm, ethylcellulose, PVP, *Escherichia* *coli*

## Abstract

Urinary catheter infections remain an issue for many patients and can complicate their health status, especially for individuals who require long-term catheterization. Catheters can be colonized by biofilm-forming bacteria resistant to the administered antibiotics. Therefore, this study aimed to investigate the efficacy of silver nanoparticles (AgNPs) stabilized with different polymeric materials generated via a one-step simple coating technique for their ability to inhibit biofilm formation on urinary catheters. AgNPs were prepared and characterized to confirm their formation and determine their size, charge, morphology, and physical stability. Screening of the antimicrobial activity of nanoparticle formulations and determining minimal inhibitory concentration (MIC) and their cytotoxicity against PC3 cells were performed. Moreover, the antibiofilm activity and efficacy of the AgNPs coated on the urinary catheters under static and flowing conditions were examined against a clinical isolate of *Escherichia coli*. The results showed that the investigated polymers could form physically stable AgNPs, especially those prepared using polyvinyl pyrrolidone (PVP) and ethyl cellulose (EC). Preliminary screening and MIC determinations suggested that the AgNPs-EC and AgNPs-PVP had superior antibacterial effects against *E. coli*. AgNPs-EC and AgNPs-PVP inhibited biofilm formation to 58.2% and 50.8% compared with AgNPs-PEG, silver nitrate solution and control samples. In addition, coating urinary catheters with AgNPs-EC and AgNPs-PVP at concentrations lower than the determined IC50 values significantly (*p* < 0.05; *t*-test) inhibited bacterial biofilm formation compared with noncoated catheters under both static and static and flowing conditions using two different types of commercial Foley urinary catheters. The data obtained in this study provide evidence that AgNP-coated EC and PVP could be useful as potential antibacterial and antibiofilm catheter coating agents to prevent the development of urinary tract infections caused by *E*. *coli*.

## 1. Introduction

Urinary catheter-related urinary tract infections contribute to many bacterial infections that affect patient health. Catheters of different types and shapes have been developed with various materials, such as silicone and latex rubber [1,2]. Despite the superiority of silicone-type catheters over latex-type, some disadvantages remain [3,4,5], such as inducing a mild degree of inflammation in the urethra, especially for the full silicone type [6]. Regarding their use, there are three main types of catheters: one-use, short-term, and Foley catheters, which are used in the long term to control urine retention in patients suffering from several disorders, e.g., prostate problems, cerebrovascular injury, and spinal cord disorders [7]. Patients receiving medications via urinary catheters require that the cannulas receive special handling and sterilization to prevent severe renal infection [7]. Different management protocols and instructions have been developed in hospitals to address this issue [5,8]. However, despite all of these measures and procedures for control, catheters remain a point for bacterial invasion and colonization. They account for many nosocomial infections in the United States and worldwide [9,10]. Such diseases could worsen the patient’s condition, leading to bacteremia, longer hospital stays, and increased costs [11]. A Foley catheter for long-term catheterization showed higher microbe and bacterial accumulation than a non-Foley (short-term) catheter [12]. Such infections can vary in presentation with mild to severe symptoms and require further investigation and control in approximately 25% of hospitalized patients undergoing urinary catheterization [13]. Bacteria such as *Staphylococcus aureus*, *Escherichia coli*, *Pseudomonas aeruginosa*, *Proteus mirabilis*, *Staphylococcus epidermidis*, *Enterococcus faecalis*, and *Klebsiella pneumoniae* could accumulate on the catheter surface and form a highly resistant biofilm [12]. The produced biofilm strongly resists administered antibiotics due to the bacterial secretion of sticky extracellular polymeric compounds, which are a mixture of polysaccharides, proteins, and DNA that can reach a thickness of 500 µm and support different bacterial cultures [14,15,16,17]. It has also been demonstrated that biofilm microbes can enter a dormant metabolic state to resist chemotherapeutics and respond differently to antibacterial agents [18,19,20]. In addition to the formation of highly resistant bacteria, the possibility of persistent catheter blockage is also great. Another study revealed that bacterial biofilm resistance is related to a diminished antibiotic effect rather than poor antibiotic penetration [7].

Furthermore, multispecies biofilms can be formed from polymicrobial infections, worsening the patient’s condition. The altered environment inside the formed biofilm with the accumulation of mineral acids and waste products is also an important issue for efficient antibacterial activity [7]. The adhesion of bacteria to the catheter surface depends on different factors, such as the type of bacteria, the cannula material, and the surface hydrophobicity of the cannula.

Different strategies have been studied to control infections related to catheter cannulation, ranging from coating the inner surface with safe biocidal materials and improving catheter design [7,21]. As natural predators of bacteria, Bacteriophages could also be used with efficient coatings to prevent antimicrobial resistance [22,23]. However, further studies are required for clinical settings. Hydrolytic enzymes have also shown superior antibacterial activity against urinary tract bacterial infections, such as Pseudomonas, Streptococcus, and Bacillus, along with multidrug-resistant strains of *S. aureus* [24,25,26]. However, the production and purification of these enzymes are challenging for their marketing, and their nature as proteins makes them susceptible to denaturation during processing [7]. Polymers impregnated in nitric oxide donors and polyzwitterions are also among the materials that show pronounced antibacterial and antifouling properties [7,27,28,29]. In addition, zwitterionic polymers form a hydration layer that affects the adhesion of bacteria and their response [30]. Antimicrobial agents added to catheters, referred to as antifouling agents [21], can act via different mechanisms, such as the slow release of biocidal or bacteriostatic agents [31], modification of the inner surface of catheters to decrease bacterial adhesion [2], and contact killing [3]. One of the most effective techniques is to disrupt the formed biofilm inside the catheter, which has shown promising results in controlling urinary tract infections. Some of these antibacterial agents have been validated by clinical trials, while others are still in the development phase.

Moreover, it was found that short-peptide antibacterials could display powerful actions against highly resistant bacteria and, hence, higher efficiency [32,33]. However, these peptide antibacterial agents exhibit lower coating properties, potential toxicity, and a high initial cost of synthesis [34]. Furthermore, their coating ability is compromised when the peptides are immobilized on a silicone catheter [35]. Silver has been a commonly used metal with a proven antibacterial activity that the FDA has approved for use in urinary catheters [5]. A study by Nandkumar described an effective and cost-saving latex catheter coated with silver oxide. The authors demonstrated active inhibition of pathogens such as *E. coli*, *P. aeruginosa,* and *Proteus mirabilis* in silver oxide, which were resistant to other antibiotics [36]. Few studies have shown the proven antibacterial and antibiofilm activity of silver nanoparticle (AgNPs) surfaces functionalized by polyurethane-coated plastic catheters against a wide range of highly resistant pathogenic strains [37,38,39]. However, silver may cause some concerns regarding its cytotoxicity after long-term exposure [40]. Formulating silver into nanoparticles (NPs) can improve its performance and penetration and decrease its administered amount while decreasing its cytotoxicity [41]. In addition, it was proven that coating AgNPs with polymers could reduce their toxicity and improve their dispersion by retarding particle aggregation [42,43]. Immobilizing metal NPs on matrix polymers (nanometal polymer hybrids) efficiently prevents NPs aggregation [44]. Moreover, these hybrids play a crucial role in NPs growth and stabilization [45].

The high surface free energy of AgNPs makes them prone to aggregation, which results in the loss of bactericidal activity [46]. Thus, the use of surface stabilizing agents for AgNPs dispersions is mandatory. Polyvinyl pyrrolidone (PVP), polyethylene glycol (PEG), and ethyl cellulose (EC) are amongst the most popular stabilizing agents for AgNPs dispersion due to their good biocompatibility and low cytotoxicity [47,48,49,50]. In addition, as previously mentioned, AgNPs stabilized with PVP and EC showed proven antibacterial properties, especially with highly resistant strains of *E. coli*. It was also found that PVP coating reduces the adherence of bacteria and device encrustation in vitro compared to uncoated polyurethane catheters [51].

From all of these aforementioned aspects, this study aimed to investigate the antibacterial and antibiofilm activity of AgNPs coated with different polymers against highly resistant pathogens accumulated in urinary catheters. In addition, the efficiency of generating these nanoformulations on the catheter surface to resist bacterial colonization via a simple-one step coating technique was studied. AgNPs were prepared to utilize the polymers PVP, PEG, and EC as reductants and coating materials. The prepared AgNPs were then characterized using UV–Vis spectroscopy to confirm their formation. Furthermore, their size, charge, morphology, and physical stability under different storage conditions were investigated. A bacteriological study was also performed to screen the antimicrobial activity and determine the minimum inhibitory concentrations (MICs) of the investigated AgNPs formulations. A cytotoxicity study was carried out using PC3 cells to ensure the biocompatibility of the synthesized formulations and to select a safe concentration for catheter coating. Moreover, the antibiofilm activity was quantitatively evaluated in 96-well microtiter plates. The antibiofilm efficacy of the AgNPs coatings under static and flowing conditions using two different types of commercially available Foley urinary catheters was also investigated.

## 2. Materials and Methods

### 2.1. Materials

Polyethylene glycol 6000 (PEG 6000) was purchased from Adwic, EL-Nasr (Cairo, Egypt). Sodium borohydride (NaBH_4_) was purchased from Sd Fine-Chem Limited (India). Silver nitrate (AgNO_3_), sodium hydroxide (NaOH), ethyl cellulose (EC), and polyvinyl pyrrolidone (PVP) were purchased from Alpha Chemicals (Cairo, Egypt).

### 2.2. Preparation of AgNPs Stabilized with Different Polymers

AgNPs-PVP and AgNPs-EC were prepared by adapting procedures from our previous study [45]. Briefly, for PVP- and EC-stabilized AgNPs, 1 mL of 1 mM NaOH was added to a stock solution of 1 mM silver nitrate under stirring, and this solution was maintained at 90–100 °C. Next, 2 mL of 1.0% PVP or EC aqueous solution was added dropwise, and the mixture stirred for another 20–30 min, then cooled to room temperature. For PEG-stabilized AgNPs, 2.0 mL of 1 mM AgNO_3_ solution was added to 48 mL of 1.0% PEG solution and stirred for 15 min in a dark place. This was followed by dropwise additions of 2 mM sodium borohydride (NaBH_4_) until the solution became yellow [52,53]. The obtained colloids were centrifuged for 15 min at 15,000 rpm, and the NPs were redispersed in deionized water and kept at 4.0 ± 0.5 °C.

### 2.3. Characterization of the Prepared AgNPs

#### 2.3.1. UV–Vis Spectroscopy

AgNP-PVP, AgNP-PEG, and AgNP-EC colloidal solutions were first centrifuged, and the obtained pellets were dispersed in distilled water and scanned from 300–600 nm using a UV–Vis spectrophotometer (Lambda 25, Perkin Elmer, Singapore).

#### 2.3.2. Size and ζ Potential

The geometric particle sizes and ζ potentials of the different AgNPs formulations were investigated using a Malvern Zetasizer Nano Z.S. (Malvern Instruments, Malvern, UK). Briefly, 1 mL of an aqueous solution of each sample was adjusted to ≈25 °C, and underwent subsequent exposure to a laser beam at ≈633 nm at a scattering angle of ≈90°. The average of three measurements was calculated, and each measurement was performed ≈20 times (with a 10 s duration).

#### 2.3.3. Particle Morphology

The shape, morphology, and size of each prepared AgNPs dispersion were visualized using transmission electron microscopy (TEM; JEOL JSM-550, Japan). Drops of each AgNPs solution were placed on the surface of the double-sided copper conductive tape and left to dry at room temperature. Then, a thin layer of platinum was coated in a vacuum chamber for 55 s at 25 mÅ using a sputter coater (JOEL JFC-1300), which facilitated their inspection and imaging.

### 2.4. Physical Stability

The physical stability of the prepared AgNPs stabilized with the investigated polymers under two different storage conditions (25.0 ± 0.5 °C and 4.0 ± 0.5 °C, 3 months each) was investigated. NPs size and charge were determined at the beginning and end of storage. Furthermore, the particle dispersions were viewed for any color change and/or the appearance of large aggregates or precipitates.

### 2.5. Microbiological Analysis

#### 2.5.1. Screening for the Antibacterial Activity of the AgNPs

The antibacterial activity of the synthesized AgNPs was investigated by the agar well diffusion method [54] using a suspension of *E. coli* (clinically isolated from patients suffering from urinary tract infections admitted to Assiut University Hospital). This *E. coli* isolate was also used for the MIC determination and antibiofilm experiments. Bacteria were inoculated on Mueller–Hinton agar plates (HiMedia Laboratories, India). Then, 5 mm wells were made using a borer. The samples to be tested were added at a 50 µg mL^−1^ concentration in these wells. Following 24 h of incubation at 37 °C, the inhibition zones were measured as the diameter in millimeters ± S.D. Experiments were repeated three times to ensure reproducibility.

#### 2.5.2. Determination of the MIC

The MICs of AgNPs-EC and AgNPs-PVP solutions were determined against clinically isolated *E. coli* using the broth microdilution method [55]. Based on CLSI’s guidelines, serial dilutions were prepared in 100 µL of Mueller–Hinton broth, to which the *E. coli* suspension was added. Following incubation for 24 h at 37 °C, the MIC was visually estimated as the NPs concentration of the well showed clear broth and no evidence of visible bacterial growth.

### 2.6. Evaluation of Antibiofilm Activity

As previously described, a quantitative evaluation of biofilm production was performed in 96-well microtiter plates [56]. NPs formulations and controls were added to wells containing the *E. coli* suspension at a 0.5 × 10^5^ CFU mL^−1^ concentration. Then, the plates were incubated at 37 °C for 24 h, and the formed biofilms were washed and stained with crystal violet (CV) stain. The absorbance values (λ_max_ at 570 nm) of the wells treated with the formulations were measured using Epoch microplate reader (Bio-Tek instruments, Winooski, VT, USA), and the results are expressed as a percentage relative to the values obtained from the control wells treated with control samples (corresponding polymeric materials).

### 2.7. Cytotoxicity Assay

The effects of various treatments of AgNPs on cell viability were evaluated using a WST-1 assay with an Abcam^®^ kit (WST-1 Cell Proliferation Reagent). PC3 cells (prostate cancer cell line, obtained from Nawah Scientific Inc., Cairo, Egypt) were plated at a density of 3 × 10^3^ cells/well in a 96-well plate and incubated in complete 10% DMEM for 24 h. Then, the cells were treated with serial dilutions of the tested formulations. After 48 h of drug–cell contact, the cells were treated with WST-1 reagent, and the absorbance was measured (FLUOstar Omega plate reader, Ortenberg, Germany) to evaluate the cellular viability and IC_50_.

### 2.8. Coating of Urinary Catheters

This study used two types of urinary catheters: silicone-coated latex Foley catheters and all silicone Foley catheters. AgNPs-PVP and AgNPs-EC at concentrations lower than the obtained IC_50_ values were used to coat urinary catheters using a process based on soaking the catheters in coating solution described previously by other researchers [57,58]. Briefly, catheter segments approximately 5 cm in length were dipped in NPs solutions at room temperature for 48 h. The catheters were washed with distilled water twice, air-dried for 24 h, and stored under ambient laboratory conditions for further investigation. It was anticipated that both catheter surfaces (internal and external) were efficiently coated after soaking in the coating solution.

### 2.9. Antibiofilm Efficacy of Urinary Catheters Coated with AgNPs under Static and Flowing Conditions

The ability of the AgNPs to inhibit biofilm formation on urinary catheters was investigated in the presence of *E. coli*. Catheters were coated with solutions containing 15 µg/mL AgNPs and incubated overnight with bacterial suspensions. Bacterial adhesion to the surface of the catheters was determined by staining 1 cm^2^ of the catheter with methyl violet and measuring the intensity of the biofilm colorimetrically as previously described. Uncoated catheters were used as controls. Moreover, the antiadhesion effects of the AgNPs-coated catheters were also tested under conditions similar to those detected in vivo. The method was adapted from [53]. Briefly, the *E. coli* suspension flowed over the surfaces of the AgNPs-coated urinary catheters using a peristaltic pump (Decdeal Ultra-quiet Mini DC 5 V, 4.8 W brushless water pump, China) for 24 h at 37 ± 0.5 °C at a rate of 20 mL/min. The catheters were washed twice with distilled water, and the bacterial adhesion to each catheter was visualized by CV staining.

### 2.10. Statistical Analysis

The Student’s *t*-test was used to determine the effects of the different N.P. treatments. The significance level was set at *p* < 0.05. All statistical analyses were performed using SPSS version 21.

## 3. Results and Discussion

### 3.1. NP Preparation

AgNPs with different polymeric materials acting as reducing agents and stabilizing capping agents were efficiently prepared, as observed from the solution color changes. The process of color change was visualized for a maximum time of approximately 30 min for all the investigated polymers. AgNPs-EC solution turned dark brown within 20 min, while AgNPs-PEG showed a characteristic yellow color within minutes after gradually adding sodium borohydride solution. Here, silver nitrate was first reduced with sodium borohydride, and the produced NPs were stabilized with PEG for AgNPs-PVP. The process of color change was noticed after 20 min via the appearance of a brown color. PVP is considered one of the most popular stabilizing materials for AgNPs dispersions due to its low cytotoxicity and good biocompatibility [46,59]. The color change in all the above-mentioned cases revealed a reduction of silver nitrate and the formation of AgNPs-capped polymers [43,47]. The NPs solutions did not show any precipitation or large aggregate formation due to the stabilizing actions of the investigated polymers. This observation was also reported in previous work by our group focusing on studying the coating of AgNPs with different types of cellulosic polymers [47] and by other researchers utilizing polymers such as chitosan, PEG, PVP, and PVA [59,60,61]. It is also worth mentioning that the investigated polymers have a negative charge that can interact with silver cations, facilitating AgNPs formation and stabilization [62,63].

### 3.2. NP Characterization

#### 3.2.1. UV–VIS Spectroscopy

Figure 1 shows the UV–VIS spectra of the different AgNPs-stabilized particles. Generally, AgNPs, with their pronounced surface plasmon resonance effect, revealed strong absorption bands in the visible region. The AgNPs-PEG, AgNPs-EC, and AgNPs-PVP spectra displayed λ_max_ values of 397, 419, and 420 nm, respectively, demonstrating the successful preparation of the NPs and the spherical morphology of the obtained particles [37,51,64]. It was reported by Pai et al. that AgNPs with a spherical morphology have only one absorption peak compared with other shapes, such as triangles and disks, both of which show more than one peak [48]. In addition, one absorption band for each type of NPs delineates the symmetrical particle distribution [65]. The observed redshift in the spectra of both the AgNPs-EC and AgNPs-PVP also revealed larger particles compared with AgNPs-PEG, as previously reported [66].

#### 3.2.2. Size and Charge

The geometric sizes and zeta potentials of the AgNPs were determined using a Zetasizer Nano. The AgNPs-EC, AgNPs-PVP, and AgNPs-PEG had geometric sizes and charges of 163.0 ± 0.9, 122.23 ± 17.61, and 79.7 ± 8.75 nm and −16.7 ± 0.2, −19.23 ± 0.61, and −20.0 ± 2.0 mV, respectively. The larger AgNPs-EC and AgNPs-PVP particles measured with dynamic light scattering (DLS) gave results that matched those from the UV–VIS spectroscopy. The AgNPs-EC were larger particles than the other polymers, possibly due to the formation of a thick coat surrounding the formed NPs. A similar result was observed in previously published work regarding utilizing different cellulosic polymers for AgNPs synthesis and stability [47]. The smaller particle size of the produced AgNPs-PEG could be attributed to the trapping of small crystals and steric hindrance, which reduce particle growth [66]. The observed zeta potentials delineated the stability of the produced colloids. AgNPs with zeta potential values higher than 10 or less than −10 are considered a stable colloidal system [67].

Moreover, the negative charge confirmed the efficient polymer coating layer around the formed AgNPs. In the case of AgNPs-PVP, adsorption of PVP onto the surface of AgNPs occurred through the carbonyl group of the PVP ring [68,69]. Garcia et al. [59] also reported a similar result. Regarding AgNPs-EC, it was reported in our research group that the negatively charged EC facilitates the attraction of Ag+ to the EC chains, followed by reduction with the present, reducing groups in the polymer chain [49]. In addition, the observed surface plasmon resonance of the produced NPs shifted to a higher wavelength than the AgNO_3_ value (the reported λ_max_, 390 nm), following other researchers [59]. This also delineates the alteration of the chemical environment around the AgNPs surface by the used polymers [70].

#### 3.2.3. Particle Morphology

The TEM images confirmed the spherical nature of the prepared NPs, as observed in their micrographes (Figure 2). The produced NPs had average particle size diameters of 54.3 ± 11.6, 21.17 ± 0.8, and 22.6 ± 8.4 nm for AgNPs-EC, AgNPs-PVP, and AgNPs-PEG, respectively. AgNPs-citrate had an average particle size of 11.2 ± 1.8 nm, as determined in our previous work [47]. Again, the AgNPs-EC (Figure 2A) was larger than the AgNPs-PVP (Figure 2B) and AgNPs-PEG (Figure 2C), according to the geometric size study. Generally, the observed particle size by TEM was lower than that obtained from DLS, as TEM shows only the metal core of the particle. At the same time, DLS measures the hydrodynamic particle size [71,72]. In addition, the coating effect of the polymers on the surface of NPs decreased the particle diffusivity, increasing the particle size measured by DLS [73].

### 3.3. Physical Stability

Visual inspection of different NPs colloidal solutions revealed the stabilities of the various prepared NPs colloids when stored under the two investigated conditions. The nanoformulations showed no precipitates or color changes over the studied time frame. Additionally, the AgNPs-PVP did not offer any significant change in size and/or charge when stored at 4.0 ± 0.5 °C for three months; however, at high temperature, the particles showed a significant difference in size (*p* < 0.05; ANOVA/Tukey). The particles showed aggregates of approximately 3–4 particles to obtain a final size of 415.66 ± 34.03 nm and a charge of −11.03 ± 2.5 mV. This observation was also recorded in our previous study [47]. In addition, lowering the zeta potential value demonstrated the formation of particle aggregates. Moreover, the AgNPs-EC showed a highly stable size and charge during storage under the two investigated conditions, as previously noted [47]. The AgNPs-PEG showed a change in size upon storage under the two different conditions. The size and charge were approximately 179.83 ± 3.57 nm and −24.33 ± 2.51 mV and 148.56 ± 8.51 nm and −21.66 ± 3.05 mV at room temperature and 4.0 ± 0.5 °C, respectively. Even though the AgNPs-PEG had a relatively large particle size after storage, their zeta potential values remained high and unchanged, which revealed the stability of the produced colloid via the steric hindrance from neutral PEG rather than the electrostatic repulsion as pointed out by other researchers [74,75].

### 3.4. Screening of the Antimicrobial Activity and Determination of MIC

AgNPs-EC and AgNPs-PVP formulations produced the largest inhibition zones (17 ± 2.0 and 19 ± 2.0, respectively). However, AgNO_3_ produced an inhibition zone of 9 ± 2 mm. AgNPs-PEG did not show a pronounced inhibition zone using the tested concentration against *E. coli* (3 ± 1 mm) (Figure 3a). Similar results were also reported using AgNPs-PEG against *E. coli* [47,76]. The larger clear zone around the AgNPs wells indicated that the nanoparticles could inhibit the growth of the *E. coli* more than the silver nitrate solution. The MIC was further determined since the agar diffusion test is a preliminary screening test for antibacterial action. The MIC values of the AgNPs-EC and AgNPs-PVP were found to be 11 ± 1.0 and 9 ± 2.0 µg/mL, respectively, while AgNO_3_ was found to have a MIC value of 17 ± 2.0 µg/mL and AgNPs-PEG showed a MIC value of 27 ± 5.0 µg/mL. The MIC values showed that *E. coli* was more susceptible to AgNPs-EC and AgNPs-PVP than silver nitrate solution. There was also a nonsignificant difference between these two AgNPs formulations (*p* > 0.05; *t*-test. Previously, it was determined that AgNPs stabilized with EC had greater antibacterial activity and a lower MIC than those stabilized with PEG and other cellulosic polymers [47]. The minimal antibacterial activity of AgNPs-PEG could be attributed to the strong binding of silver cations to the NPs, as described before [76], or due to forming a thick coat around the AgNPs. Similarly, it was reported that AgNPs stabilized with PVP had higher antibacterial activity against *E. coli*, *S. aureus*, and *Candida albicans* when the particles were prepared using polymer concentrations up to 0.5% [59].

### 3.5. Cell Toxicity

The cytotoxic effects of AgNPs-EC and AgNPs-PVP on PC3 cells were investigated using a WST-1 assay. Our results showed that the IC_50_ values of the AgNPs-EC and AgNPs-PVP were 18.49 µg/mL and 16.81 µg/mL, respectively. Interestingly, the synthesized NPs exerted marked antibacterial effects at lower concentrations, indicating their biocompatibility and safe use for further investigations and coatings.

### 3.6. Evaluation of Antibiofilm Activity

Ideally, the catheter coating material should possess inhibitory activity against urinary pathogens. However, the frequent use of antibiotic coating may develop resistance to these antibiotics. Another challenge is the wide variety of infectious agents that may contaminate the catheters, which makes it difficult to choose a certain coating antibiotic. In addition, there is also a potential risk of causing side effects to patients [77]. On the other hand, AgNPs are among the most popular coating material for urinary catheters. Silver was proven to harm the bacteria by disrupting the cell membrane and proteins and stimulating the release of reactive oxygen species and oxidative stress by releasing Ag ions into the urinary bladder [7].

The effect of these AgNPs on *E. coli* biofilm formation was tested with a microtiter plate assay. The results showed that treatment of cells with subinhibitory concentrations of AgNPs (5 µg mL^−1^) significantly reduced biofilm formation in most cases. As shown in Figure 3b, AgNPs-EC treatment reduced biofilm formation to 58.2% (*p*-value = 0.01) compared to untreated control wells (100%). In contrast, AgNPs-PVP and biofilm inhibition were 50.8±% (*p*-value = 0.01), indicating that both nanoformulations had potent antibiofilm activities. Samples treated with AgNO_3_ and AgNPs-PEG revealed biofilm formation that reached 75.6% (*p*-value = 0.02) and 85.5% (*p*-value = 0.07), respectively, compared to the untreated control samples.

Moreover, as shown in Figure 4a,b, catheters coated with AgNPs significantly (*p* < 0.05; *t*-test) inhibited biofilm development compared to the noncoated control catheters under static conditions. Additionally, a clear nonsignificant (*p* > 0.05; *t*-test) difference was found for the two investigated types of AgNPs and the two types of urinary Foley catheters used. In addition, we used a pump to allow a bacterial suspension to flow through the coated catheters to mimic in vivo conditions and tested the ability of the bacteria to form biofilms on these coated catheters. A trend similar to the static state was found using the flowing conditions. It is also worth mentioning that there was a significant (*p* < 0.05; *t*-test) reduction in adhesive capability relative to the noncoated catheters (Figure 4c,d) using a pump at a rate of 20 mL min^−1^. The AgNPs-EC- and AgNPs-PVP-coated catheters showed biofilm formation inhibition values of 49.3 ± 10.2% and 41.1 ± 4.1% for the latex Foley catheters and 54.7 ± 13.1% and 56.2 ± 12.6% for the all-silicone Foley catheters, respectively. Similar results were also obtained by other researchers who tried to control biofilm formation related to *E. coli* and *S. aureus* on urinary silicone catheters coated with poly(DMA-mPEGMA-AA) under flowing conditions [53]. These results indicated that coating catheters with a simple one-step technique could inhibit bacterial adhesion and biofilm development on the catheter surface and eventually control urinary catheter-related infections.

## 4. Conclusions

In this study, it was proven that different polymeric materials have the potential to form stabilized colloidal dispersions of AgNPs when used at a concentration of 1% *w*/*v*. The observed negative zeta potentials for the AgNPs formulations indicated the efficient coating of the polymers used around the AgNPs. In addition, the TEM images revealed the spherical character of the AgNPs-stabilized polymers. AgNPs-EC and AgNPs-PVP showed higher physical stability towards aggregation than AgNPs-PEG when stored at low temperatures. Both AgNPs-PVP and AgNPs-EC have higher antibacterial activity against Gram-negative *E. coli* than silver nitrate and AgNPs-PEG. Treatment of cells with subinhibitory concentrations of AgNPs (5 µg mL^−1^) significantly reduced biofilm formation. In addition, commercial Foley urinary catheters were coated with the nanoformulations via a simple one-step method and investigated for their ability to inhibit bacterial adhesion and subsequent biofilm formation. AgNPs-PVP- and AgNPs-EC-coated catheters inhibited biofilm development compared to noncoated catheters. In addition, bacterial suspensions flowing through the catheters for 24 h showed reduced adhesive capabilities relative to the noncoated catheters, similar to the results of the static conditions. The obtained data demonstrated the high potential applicability of coating Foley catheters (silicone and latex-coated silicone) with AgNPs-EC or AgNPs-PVP to inhibit biofilm formation and, eventually, urinary tract-related catheter infections. Thus, the Foley catheter postmodification coating process with the safest concentrations of AgNPs-PVP or AgNPs-EC could improve their resistance to life-threatening pathogenic infections.

## Figures and Tables

**Figure 1 microorganisms-10-01297-f001:**
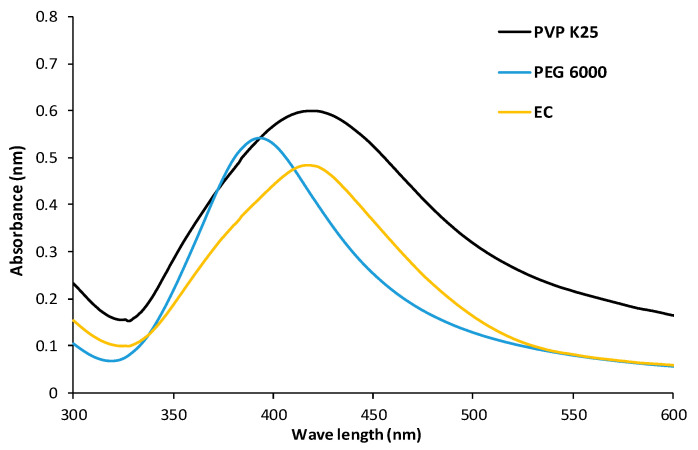
UV–VIS spectra for AgNPs-PVP, AgNPs-PEG, and AgNPs-EC.

**Figure 2 microorganisms-10-01297-f002:**
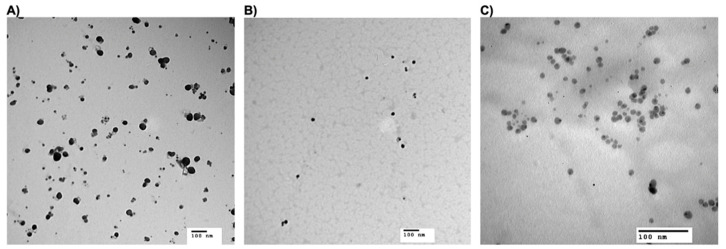
Transmission electron micrographs for AgNPs-EC (**A**), AgNPs-PVP (**B**), and AgNPs-PEG (**C**). The scale bar represents 100 nm.

**Figure 3 microorganisms-10-01297-f003:**
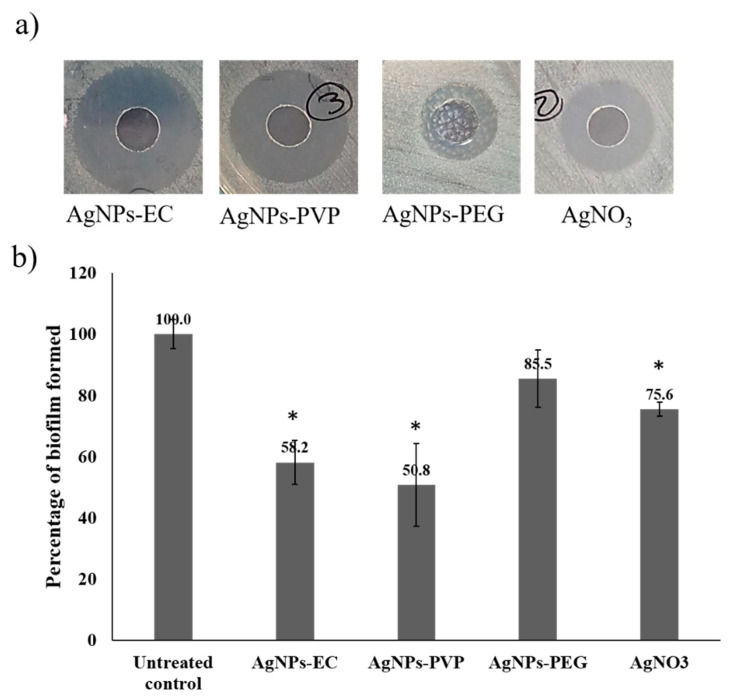
Antibacterial and antibiofilm activities of the differently prepared AgNPs-stabilized polymers and silver nitrate solution. (**a**) Inhibition zone diameters induced by the formulations and (**b**) biofilm formation in the presence of different tested samples. Columns show the mean values of 3 experiments ± S.D. * Denotes a *p*-value < 0.05 as tested by unpaired *t*-test compared with the untreated control samples.

**Figure 4 microorganisms-10-01297-f004:**
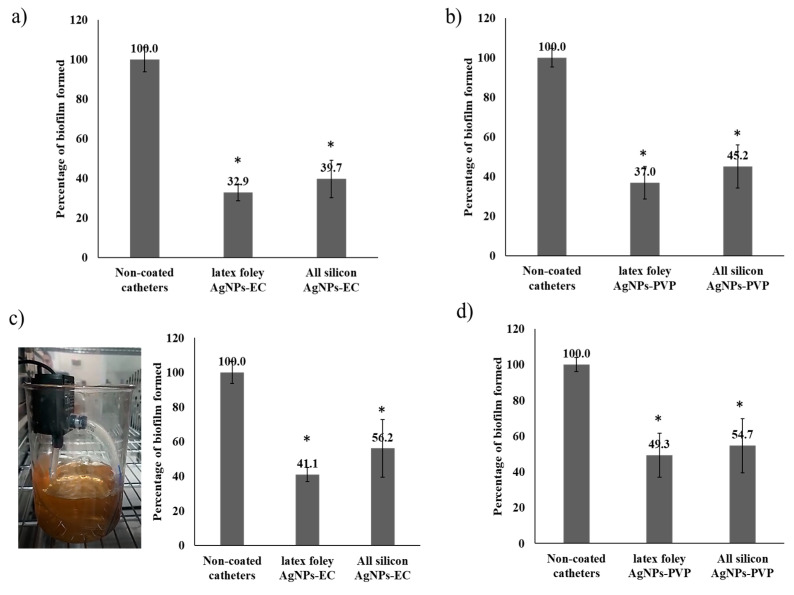
Effect of catheter coating on biofilm formation. (**a**) The percentage of biofilm formation on catheters coated with AgNPs-EC under static conditions. (**b**) Biofilms formed on catheters coated with AgNPs-PVP under static conditions. Panels (**c**,**d**) show the effect of surface coating on biofilm formation under flowing conditions. Columns show the mean values of 3 experiments ± S.D. * Denotes *p*-value < 0.05 as tested by unpaired *t*-test compared with the untreated control samples. The image on trace (**c**) shows the flowing conditions with a pump rate of 20 mL min^−1^. Two hundred ml of *E. coli* culture suspension was stirred to prevent bacterial sedimentation, and the experiment was conducted in an incubator at a temperature of 37 ± 0.5 °C.

## Data Availability

Data is contained within the article.

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
