# Peer review of "Retardation of Bacterial Biofilm Formation by Coating Urinary Catheters with Metal Nanoparticle-Stabilized Polymers"

_microorganisms, 2022, doi:10.3390/microorganisms10071297_

Round 1

Reviewer 1 Report

I have read the manuscript entitled "Retardation of Bacterial Biofilm Formation by Coating Urinary Catheters with Metal Nanoparticle-Stabilized Polymers" with great interest and I think it is in principle suited for a publication in the Microorganisms. The authors aim to help solve the problem of the urinary catheter infections by developing new catheter coating agents with the antimicrobial metal nanoparticle-stabilized polymers. The manuscript brings some new information to the scientific community; nevertheless, I have some concerns. For the microbiological analysis no antibiotics (e.g. chloramphenicol, gentamicin sulfate), hydrolytic enzymes, or antimicrobial peptides were used as positive controls. Please provide further explanation to support your polymers and what advantages they have given over other approaches.

Minor comments: 

Please check the section "Author Contributions".

Also, please carefully check and correct abbreviations (e.g. D.N.A., M.I.C. remove dots).

Author Response

June 21, 2022

Professor Martin Von Bergen,

Editor-in-chief (Microorganisms)

Dear Professor Von Bergen,

On behalf of my colleagues, it is my pleasure to resubmit the revised original article manuscript entitled: “Retardation of Bacterial Biofilm Formation by Coating Urinary Catheters with Metal Nanoparticle-Stabilized Polymers” for potential consideration in Microorganisms.

Reviewer I

I have read the manuscript entitled "Retardation of Bacterial Biofilm Formation by Coating Urinary Catheters with Metal Nanoparticle-Stabilized Polymers" with great interest and I think it is in principle suited for a publication in the Microorganisms. The authors aim to help solve the problem of the urinary catheter infections by developing new catheter coating agents with the antimicrobial metal nanoparticle-stabilized polymers. The manuscript brings some new information to the scientific community; nevertheless, I have some concerns.

Thank you very much for your revision and valuable comments.

For the microbiological analysis, no antibiotics (e.g., chloramphenicol, gentamicin sulfate), hydrolytic enzymes, or antimicrobial peptides were used as positive controls. Please provide further explanation to support your polymers and what advantages they have given over other approaches.

Reply: Thank you; actually, AgNO3 and levofloxacin were used as controls in our experiments. However, we preferred to show the results of AgNO3 to avoid confusing the readers and because one of the major aims of the work was to compare the efficacy of the nano-preparations (AgNPs-PVP, AgNPs-PEG, and AgNPs-EC) to the crude AgNO3, as shown in Figure 3 in the manuscript. Testing of other anti-microbial agents and the value of combing these anti-microbial agents with silver nanoparticles are to be shown in the next work from our group. In response to the reviewer comment, we added the following paragraph that supports our results:

Ideally, the catheter coating material should possess inhibitory activity against urinary pathogens. However, the frequent use of antibiotic coating may develop resistance to these antibiotics. Another challenge is the wide variety of infectious agents that may contaminate the catheters, making it difficult to choose a certain coating antibiotic. In addition, there is also a potential risk of causing side effects to patients (1). On the other hand, silver nanoparticles are among the most popular coating material for urinary catheters. Silver was proven to harm the bacteria by disrupting the cell membrane and proteins and stimulating the release of reactive oxygen species and oxidative stress by releasing Ag ions into the urinary bladder (2).

Regarding the used polymers in our study and their advantages over other approaches.

We have shown in the introduction section some of the materials used as anti-fouling agents like antimicrobial peptides. Even though they showed a pronounced efficiency, they still have concerns regarding lower coating capacity, toxicity, and the high initial cost of synthesis (3). Highlighted in yellow color in the introduction

This is why looking for other materials is considered an important issue. Utilizing polymers like EC and PVP demonstrated many advantages in terms of lowering the cytotoxicity of silver nanoparticles, stabilizing the nanoparticles' dispersion, and decreasing the potential aggregation of the produced NPs (4, 5). This is also highlighted in the introduction

The high surface free energy of silver nanoparticles makes them prone to aggregation, resulting in the loss of bactericidal activity (6). Thus the use of surface stabilizing agents for silver nanoparticles dispersions is mandatory.

PVP, PEG, and EC are the most popular stabilizing agents for AgNPs dispersion due to their good biocompatibility and low cytotoxicity (6-9). In addition, as previously mentioned, AgNPs stabilized with these polymers showed proven antibacterial properties, especially with highly resistant strains of E. coli (10, 11). EC showed high antibacterial activity and superior physical stability (11). It was also found that PVP coating reduces the adherence of bacteria and device encrustation in vitro compared to uncoated polyurethane catheters (12).

From all of these aspects mentioned above, we are investigating in this work the utilization of these polymers as stabilizing agents for AgNPs and their effect on inhibiting biofilm formation in urinary catheters.

Please see lines (112-115), Lines (131-139)

Please see Lines (381-389)

Minor Comments:

Please check the section "Author contributions"

  • The author contributions section has been checked and added to the manuscript.

Also, please carefully and correct abbreviations (e.g. D.N.A., M.I.C. remove dots)

  • Abbreviations have been corrected all over the manuscript and checked

References Rev. I:

  • Marissa J. Andersen, Ana L. Flores-Mireles. Urinary catheter coating modifications: The race against catheter associated unfections. Coatings 2020, 10(1), 23; https://doi.org/10.3390/coatings10010023
  • Priyadarshini Singha, Jason Locklin, Hitesh Handa. A review of the recent advacnes in antimicrobial coatings for urainary cathters. Acta Biomaterialia Volume 50, 1 March 2017, Pages 20-40, https://doi.org/10.1016/j.actbio.2016.11.070
  • Kim, J.S.; Kuk, E.; Yu, K.N.; Kim, J.H.; Park, S.J.; Lee, H.J.; Kim, S.H.; Park, Y.K.; Park, Y.H.; Hwang, C.Y.; et al. Antimicrobial effects of silver nanoparticles. Nanomedicine 2007, 3, 95-101, doi:10.1016/j.nano.2006.12.001.
  • Capadona, J.R.; Van Den Berg, O.; Capadona, L.A.; Schroeter, M.; Rowan, S.J.; Tyler, D.J.; Weder, C. A versatile approach for the processing of polymer nanocomposites with self-assembled nanofibre templates. Nat Nanotechnol 2007, 2, 765-769, doi:10.1038/nnano.2007.379.
  • Wang, L.S.; Wang, C.Y.; Yang, C.H.; Hsieh, C.L.; Chen, S.Y.; Shen, C.Y.; Wang, J.J.; Huang, K.S. Synthesis and anti-fungal effect of silver nanoparticles-chitosan composite particles. Int J Nanomedicine 2015, 10, 2685-2696, doi:10.2147/IJN.S77410.

  • Shrivastava S, Bera T, Roy A, Singh G, Ramachandrarao P, Dash D (2007) Characterization of enhanced antibacterial effects of novel silver nanoparticles. Nanotechnology 18:225103. https://doi.org/10.1088/0957-4484/18/22/225103
  • Pal S, Tak YK, Song JM (2007) Does the antibacterial activity of silver nanoparticles depend on the shape of the nanoparticle? A study of the gram-negative bacterium Escherichia coli. Appl Environ Microbiol 73:1712–1720. https://doi.org/10.1128/AEM.02218-06
  • Ahmed A. H. Abdellatif, Mansour Alsharidah, Osamah Al Rugaie, Nahla Sameh Tolba, Hesham M. Tawfeek. Silver Nanoparticles Coated Ethyl Cellulose Inhibits Tumour Necrosis Factor-α of Breast Cancer Cells. Drug design, Development and Therapy 2021:15, 2035-2046. https://doi.org/10.2147%2FDDDT.S310760
  • Ahmed Abdelfattah, Ahmed E. Aboutaleb, Abu‐Baker M Abdel‐Aal, Ahmed A. H. Abdellatif, Hesham M Tawfeek,Sayed I. Abdel-Rahman. Design and optimization of PEGylated silver nanoparticles for efficient delivery of Doxorubicin to cancer cells. Journal of Drug Delivery Science and Technology Volume 71, May 2022, 103347, https://doi.org/10.1016/j.jddst.2022.103347.
  • Rayna Bryaskova, Daniela Pencheva, Stanislav Nikolov, Todor Kantardjiev. Synthesis and comparative study on the antimicrobial activity of hybrid materails based on silver nanoparticles (AgNps) stabilized by polyvinylpyrrolidone (PVP). Journal of Chemical Biology 4, 185, 2011.
  • Ahmed A.H. Abdelftah, Hamad Al-Turki, Hesham M. Tawfeek. Different cellulosic polymers for synthesizing of silver nanoparticles with antioxidant and antibacterial activities. Scientific Reports-Nature 11, 84, 2021. https://doi.org/10.1038/s41598-020-79834-6
  • Tunney MM, Gorman SP. Evaluation of a poly(vinyl pyrollidone)-coated biomaterial for urological use. Biomaterials 23, 4601-4608, 2002, https://doi.org/10.1016/S0142-9612(02)00206-5

Sincerely yours,

Osamah Al Rugaie

Department of Basic Medical Sciences, College of Medicine and Medical Sciences, Qassim University, Unaizah, P.O. Box 991, Al Qassim 51911, Saudi Arabia; Email: [email protected]

Hesham M. Tawfeek,

Professor and chairman of Industrial Pharmacy Department

Faculty of Pharmacy, Assiut University, Assiut 71526, Egypt

 Tel: 00201094246149; E-mail: [email protected]

Reviewer 2 Report

Manuscript reports on preparation of Ag nanoparticles wrapped out in two polymers - polyvinyl pyrrolidone and ethyl cellulose, their spectroscopic and physical characteristics, morphology and interactions with bacterial strains of various genera in gel suspension and in biofilm form. Data are nicely supplemented with studies on usefulness of such NP’s to reduce bacterial flora in clinically used urinary catheters. Manuscript is continuation of Authors work on AgNp’s externally functionalized with polymeric materials and their application in medicine and in general uses similar well developed techniques. And has some shortages that are obvious to the authors but not necessarily to the readers. It is an interesting paper from scientific and practical perspective and should attract many readers. It might be published in Microorganisms as a full paper after considering few changes shown below.

1. Line 152: “polyvinyl pyrrolidone (PVP) and ethyl cellulose …”; what amount of the polymer in relation to AgNO3 were added?

2.Line 263: “The color change in all the cases mentioned above revealed the complete reduction of silver nitrate…” Statement about completenes of the process is not true since AgNO3 is colorless. By the way it would be interesting to analyse spectroscopically supernanat solution after centrifugation or at least weight dry deposit, if any.

3.Line 269:”It is also worth mentioning that the investigated polymers have a negative charge that can interact with silver cations, facilitating AgNP formation and stabilization [57,58].” Mentioned in ref 58 carboxymethyl cellulose is negtively charged due to carboxy group. Explain on chemical background why NP’s are negatively charged

4.Line 304: “...particle size diameters of 54.3±11.6, 21.17±0.8, and 22.6±8.4 nm for AgNPs-EC, AgNPs-PVP,and AgNPs-PEG, respectively.…”. - add mm.

5.Both discussion covered by chapter 3.8. Evaluation of antibiofilm activity and caption under Fig.3 and Fig. 4 need information about bacterial strain used for biofilm formation.

6.Line 409: Sentence “In addition, the T.E.M. images revealed the spherical character of the AgNP-stabilized polymers with a smaller size than that observed inhe D.L.S.” should note given in conclusions since the difference between the two techniques is obvious and well discussed in literature.

Summing up minor revision is suggested.

Author Response

June 21, 2022

Professor Martin Von Bergen,

Editor-in-chief (Microorganisms)

Dear Professor Von Bergen,

On behalf of my colleagues, it is my pleasure to resubmit the revised original article manuscript entitled: “Retardation of Bacterial Biofilm Formation by Coating Urinary Catheters with Metal Nanoparticle-Stabilized Polymers” for potential consideration in Microorganisms.

Reviewer II.

Manuscript reports on preparation of Ag nanoparticles wrapped out in two polymers - polyvinyl pyrrolidone and ethyl cellulose, their spectroscopic and physical characteristics, morphology and interactions with bacterial strains of various genera in gel suspension and in biofilm form. Data are nicely supplemented with studies on usefulness of such NP’s to reduce bacterial flora in clinically used urinary catheters. Manuscript is continuation of Authors work on AgNp’s externally functionalized with polymeric materials and their application in medicine and in general uses similar well developed techniques. And has some shortages that are obvious to the authors but not necessarily to the readers. It is an interesting paper from scientific and practical perspective and should attract many readers. It might be published in Microorganisms as a full paper after considering few changes shown below.

Thank you very much for your revision and valuable comments.

  1. Line 152: “polyvinyl pyrrolidone (PVP) and ethyl cellulose …”; what amount of the polymer in relation to AgNO3 were added?

PVP and EC were used in 1% wlv (1g/100ml) and AgNO3 was used in 1mMole (17mg/100ml).

Similar ratios for polymer-based AgNPs stabilization were also reported (1-3).

The used concentration of these polymers is also within the reported optimum concentration for high antimicrobial activity. It was also stated that PVP concentration up to 5.0% demonstrated an optimum antibacterial activity (4).

2.Line 263: “The color change in all the cases mentioned above revealed the complete reduction of silver nitrate…” Statement about completenes of the process is not true since AgNO3 is colorless. By the way it would be interesting to analyse spectroscopically supernanat solution after centrifugation or at least weight dry deposit, if any.

Thank you very much for your comments. We agree with you as we discussed with all authors; the word complete is not appropriate; it was deleted.

Please see Lines (274-275)

3.Line 269:”It is also worth mentioning that the investigated polymers have a negative charge that can interact with silver cations, facilitating AgNP formation and stabilization [57,58].” Mentioned in ref 58 carboxymethyl cellulose is negtively charged due to carboxy group. Explain on chemical background why NP’s are negatively charged

Thanks for this comment, In the case of AgNPs-PVP, adsorption of PVP onto the surface of AgNPs occurred through the carbonyl group of the PVP ring (5, 6). Garcia et al., 2020 (4) also reported a similar result. Regarding AgNPs-EC, it was reported in our research group that the negatively charged EC facilitates the attraction of Ag+ to the EC chains. Followed by reduction with the present, reducing groups in the polymer chain (1). In addition, the observed surface plasmon resonance of the produced NPs shifted to a higher wavelength than the AgNO3 value (the reported λmax, 390 nm), which is in accordance with other researchers (4). This also delineates the alteration of the chemical environment around the AgNPs surface by the used polymers (7).

Please see lines (312-320)

4.Line 304: “...particle size diameters of 54.3±11.6, 21.17±0.8, and 22.6±8.4 nm for AgNPs-EC, AgNPs-PVP,and AgNPs-PEG, respectively.…”. - add mm.

The calculated particles size using the DLS technique in nanometers, not in millimeters

5.Both discussion covered by chapter 3.8. Evaluation of antibiofilm activity and caption under Fig.3 and Fig. 4 need information about bacterial strain used for biofilm formation.

Reply: Thank you. In order to provide reliable results, we preferred to carry out the experiments using a clinical strain isolated from patients with urinary tract infections. We added this information to the material and methods section.

“The antibacterial activity of the synthesized AgNPs was investigated by the agar well diffusion method [48] using a suspension of E. coli (clinically isolated from patients suffering from urinary tract infections admitted to Assiut University Hospital). This E coli isolate was also used for the MIC determination and antibiofilm experiments.”

Please see lines (200-203)

6.Line 409: Sentence “In addition, the T.E.M. images revealed the spherical character of the AgNP-stabilized polymers with a smaller size than that observed inhe D.L.S.” should note given in conclusions since the difference between the two techniques is obvious and well discussed in literature.

The sentence has been removed from the conclusion section

References Rev. II:

  • Ahmed A. H. Abdellatif, Mansour Alsharidah, Osamah Al Rugaie, Nahla Sameh Tolba, Hesham M. Tawfeek. Silver Nanoparticles Coated Ethyl Cellulose Inhibits Tumour Necrosis Factor-α of Breast Cancer Cells. Drug design, Development and Therapy 2021:15, 2035-2046. https://doi.org/10.2147%2FDDDT.S310760
  • Ahmed Abdelfattah, Ahmed E. Aboutaleb, Abu‐Baker M Abdel‐Aal, Ahmed A. H. Abdellatif, Hesham M Tawfeek,Sayed I. Abdel-Rahman. Design and optimization of PEGylated silver nanoparticles for efficient delivery of Doxorubicin to cancer cells. Journal of Drug Delivery Science and Technology Volume 71, May 2022, 103347, https://doi.org/10.1016/j.jddst.2022.103347
  • Ahmed A.H. Abdelftah, Hamad Al-Turki, Hesham M. Tawfeek. Different cellulosic polymers for synthesizing of silver nanoparticles with antioxidant and antibacterial activities. Scientific Reports-Nature 11, 84, 2021. https://doi.org/10.1038/s41598-020-79834-6
  • Arthur M. Garcia, Marcos A. Bizeto, vitor B. Ferrari, Debora N. Okamoto, Suzan Pantaroto de Vasconcellos, Fernando F. Camilo. Direct evaluation of microbial growth of silver nanoparticles stabilized by poly(vinyl pyrrolidone) and poly(vinyl alcohol). J. Nanoparticles Res., 22, 137 (2020).
  • Mdluli PS, Sosibo NM, Mashazi PN, Nyokong T, Tshikhudo RT, Skepu A, van der Lingen E (2011) Selective adsorption of PVP on the surface of silver nanoparticles: a molecular dynamics study. J Mol Struct 1004:131–137.https://doi.org/10.1016/j.molstruc.2011.07.049
  • Kyrychenko A, Korsun OM, Gubin II, Kovalenko SM, Kalugin ON (2015) Atomistic simulations of coating of Silver nanoparticles with poly(vinylpyrrolidone) oligomers: effect of oligomer chain length. J Phys Chem C 119:7888–7899. https://doi.org/10.1021/jp510369a
  • Mulfinger L, Solomon SD, Bahadory M, Jeyarajasingam AV, Rutkowsky SA, Boritz C (2007) Synthesis and study ofSilver nanoparticles. J Chem Educ 84:322. https://doi.org/10.1021/ed084p322

Sincerely yours,

Osamah Al Rugaie

Department of Basic Medical Sciences, College of Medicine and Medical Sciences, Qassim University, Unaizah, P.O. Box 991, Al Qassim 51911, Saudi Arabia; Email: [email protected]

Hesham M. Tawfeek,

Professor and chairman of Industrial Pharmacy Department

Faculty of Pharmacy, Assiut University, Assiut 71526, Egypt

 Tel: 00201094246149; E-mail: [email protected]
